# Effect of *Lactococcus lactis* Strain Plasma on HHV-6 and HHV-7 Shedding in Saliva: A Prospective Observational Study

**DOI:** 10.3390/microorganisms9081683

**Published:** 2021-08-08

**Authors:** Hiroki Miura, Masaru Ihira, Kei Kozawa, Yoshiki Kawamura, Yuki Higashimoto, Fumihiko Hattori, Tetsushi Yoshikawa

**Affiliations:** 1Department of Pediatrics, Fujita Health University School of Medicine, Toyoake 470-1192, Japan; studyhardcustom@yahoo.co.jp (K.K.); yoshiki@fujita-hu.ac.jp (Y.K.); light_my_fire_54_0127@yahoo.co.jp (F.H.); tetsushi@fujita-hu.ac.jp (T.Y.); 2Faculty of Clinical Engineering, Fujita Health University School of Health Sciences, Toyoake 470-1192, Japan; mihira@fujita-hu.ac.jp; 3Faculty of Medical Technology, Fujita Health University School of Health Sciences, Toyoake 470-1192, Japan; yhigashi@fujita-hu.ac.jp

**Keywords:** HHV-6, HHV-7, saliva, LC-Plasma

## Abstract

HHV-6 and HHV-7 can reactivate in the salivary gland in response to various host stresses. *Lactococcus lactis* strain Plasma (LC-Plasma) can activate plasmacytoid dendritic cells (pDCs) and decrease viral infection. We investigated whether LC-Plasma intake could decrease HHV-6 and HHV-7 reactivation in the salivary gland. A total of 54 healthy volunteers were enrolled in this study. Participants took LC-Plasma granules daily for 6 weeks. Saliva samples were collected from subjects weekly for 4 weeks before (first), during (second), and after (third period) LC-Plasma intake. There was a 2-week interval between the first and second periods and a 3-week interval between the second and third periods. Mean salivary HHV-6 and HHV-7 DNA loads were compared among the three observation periods. In the first period (baseline data of viral DNA shedding), HHV-6 DNA shedding was significantly higher in subjects under 40 years old, and HHV-7 DNA shedding was significantly higher in males. HHV-6 and HHV-7 DNA loads did not significantly differ between periods. Meanwhile, in a subgroup analysis of the subjects under 40 years old, HHV-6 DNA load was significantly lower in the second period than in the first period. LC-Plasma decreases HHV-6 reactivation in the salivary glands in younger adults.

## 1. Introduction

Human herpesvirus 6 (HHV-6) and HHV-7 are members of Betaherpesvirinae. HHV-6 or HHV-7 primary infection can cause exanthem subitum, a common febrile exanthematous illness in early childhood [1,2,3]. These viruses can establish latency in various body sites, such as the salivary gland [4] and central nervous system [5] after primary infection [6]. Frequent viral shedding in saliva has been demonstrated, even in healthy and immunocompetent seropositive adults [7,8], and the dynamics of viral shedding may be associated with underlying disorders such as collagen diseases [9,10,11,12]. Interestingly, HHV-7 is frequently isolated from saliva, but HHV-6 is not, despite the presence of a high copy number of viral DNA [13]. Furthermore, HHV-6 and HHV-7 DNA loads in saliva are reliable biomarkers of physiological fatigue [14]. These findings suggest that HHV-6 and HHV-7 can be reactivated in the salivary gland by the various types of host stresses.

*Lactococcus lactis* subsp. *lactis* JCM 5805 is known as *Lactococcus lactis* strain Plasma (LC-Plasma). Previous in vitro and in vivo experiments showed that LC-Plasma can activate plasmacytoid dendritic cells (pDCs) [15]. pDCs are responsible for initiating adaptive immune responses, including secretion of type I interferon and antigen presentation [16,17]. Mouse experiments showed that LC-Plasma intake suppresses respiratory [18], mucosal [19], and systemic viral infections [20]. In addition, health economic analysis has shown that LC-Plasma confers economic benefits by improving worker productivity [21]. Moreover, LC-Plasma decreases fatigue accumulation during high-intensity exercise in male athletes [22].

On the basis of these findings, we hypothesized that LC-Plasma intake could decrease reactivation of HHV-6 and HHV-7 in the salivary gland by activating host immune responses and decreasing fatigue. To test this hypothesis, we monitored salivary shedding of HHV-6 and HHV-7 DNA by real-time polymerase chain reaction (PCR) before and after LC-Plasma intake.

## 2. Materials and Methods

### 2.1. Study Design

Healthy Japanese adult volunteers of both sexes were enrolled in this prospective study after consenting to participate. No subject took any supplement. A flowchart of the study design is shown in Figure 1. First, saliva samples were serially collected from the subjects once a week for 4 weeks (first period) to measure baseline viral DNA shedding. One subject became pregnant and was excluded after the first period. Next, subjects took in one pack of LC-Plasma granules (LC-Plasma) anytime daily for 6 weeks. LC-Plasma granules consist of dry powder containing approximately 100 billion cells of heat-killed LC-Plasma (Kirin Holdings Co., Ltd., Tokyo, Japan). Two weeks after starting LC-Plasma intake, which was required for the effect of LC-Plasma to be expressed [22,23], saliva samples were serially collected from the subjects once a week for four weeks (second period). After a 3-week washout period, saliva samples were serially collected from subjects once a week for 4 weeks; the subjects did not consume LC-Plasma during this time (third period). One or more milliliters of saliva collected from each subject was used to measure HHV-6 and HHV-7 DNA loads by real-time PCR. All saliva samples were stored at −20 °C until DNA extraction. Subjects were not allowed to change their eating habits or take any other supplements during the observation period. Subjects lacking two or more saliva samples each period, as well as those lacking LC-Plasma intakes for two or more consecutive days or a total of 5 or more days, were excluded from the analysis. Furthermore, subjects with no viral DNA shedding during the first period were excluded from viral DNA analyses. 

Mean copy numbers of HHV-6 and HHV-7 DNA from the four saliva samples obtained before taking LC-Plasma (first period) were calculated as baseline levels of salivary viral DNA shedding. The association was examined between salivary viral DNA shedding and sex, age, and whether the subject worked night shifts. Then, the effect of LC-Plasma on salivary HHV-6 and HHV-7 DNA shedding was evaluated. Mean copy numbers of HHV-6 and HHV-7 DNA were calculated in four consecutive saliva samples during the second and third periods, and mean viral DNA loads were compared among the three different sampling periods. Viral DNA loads of samples that were below the detection limit of real-time PCR were defined as 0 copies/mL.

### 2.2. DNA Extraction and Real-Time PCR

Saliva samples were centrifuged for 5 min at 2000 g, and then DNA samples were extracted from 200 µL of saliva supernatant using the QIAamp Blood Kit (QIAGEN, Hilden, Germany). Extracted DNAs were stored at −20 °C. Real-time PCR was used to measure copy numbers of HHV-6 and HHV-7 DNA. The primers for detecting HHV-6 DNA were selected from the U31 gene (large tegument protein). The forward primer was 5’-TTTGCAGTCATCACGAT-CGG-3’, and the reverse primer was 5’-AGAGCGACAAATTGGAGGTTTC-3’. The fluorogenic probe (5’-AGCCACAGCAGCCATCTACATCTGTCAA-3’) was located between the primers. Our HHV-6 real-time PCR could not discriminate between HHV-6A and B. However, excreted HHV-6 DNA in saliva was considered to be HHV-6B based on previous studies [24,25]. The primers for detecting HHV-7 DNA were selected from the U31 gene (large tegument protein). The forward primer was 5’-AAAGAATGGTTTTGTTCAACTCCAA-3’, and the reverse primer was 5’-ACATTCACTTTGCGTGCATTTTC-3’. The fluorogenic probe (5’-[FAM]-TCATCGAGAACATAGGAGAAGCTCCAGCA-[TAMRA]-3’) was located between the primers. The PCR reaction was carried out using a TaqMan PCR kit (PE Applied Biosystems, Foster City, CA, USA). The Z29 HHV-6B strain and the RK HHV-7 strain were used as each viral positive control, and standard curves were constructed as previously described [12,26]. As an internal standard, the copy number of the 18S rRNA gene was measured in both HHV-6 and HHV-7 negative samples by using Eukaryotic 18S rRNA Endogenous Control kit (Thermo Fisher Scientific, Waltham, MA, USA), and it was confirmed that a sufficient number of copies were detected in all of the samples.

### 2.3. Statistical Analysis

The association between HHV-6 and HHV-7 DNA loads during the first period and three factors (sex, age, and night shifts) was analyzed using the Mann–Whitney U test. To evaluate the effect of LC-Plasma on salivary HHV-6 and HHV-7 DNA shedding, we compared viral DNA loads among the first, second, and third periods using the Wilcoxon signed-rank test. *P* < 0.05 was defined as statistically significant. Statistical analyses were conducted using JMP version 13.2.1 (SAS Institute, Cary, NC, USA).

### 2.4. Ethics Approval

This study was approved by the Ethical Review Board of Human Studies at Fujita Health University (Accession number HM18-487, approved on 19 June 2019). We obtained written informed consent from all volunteers before the trial.

## 3. Results

### 3.1. Study Subjects and Collected Samples

A total of 54 healthy adults were enrolled in this study (27 males and 27 females). The median age of both the male subjects (range, 22–58 years) female subjects (range, 21–59 years) was 40 years. Job types were as follows: 26 physicians, 20 medical staff, 5 medical students, and 3 office workers. Of the 26 physicians, 20 had one to six night shifts during each period. All 54 subjects were enrolled in the baseline analysis (first period). A total of 578 saliva samples were collected from the subjects and examined by real-time PCR. For the following reasons, eleven and seven subjects were excluded from the analyses of the effect of LC-Plasma for HHV-6 and HHV-7 DNA shedding, respectively: saliva samples were not obtained serially, 5; pregnancy, 1 (HHV-6 and HHV-7); no viral DNA shedding during the first period, 5 (HHV-6) and 1 (HHV-7). Finally, 43 and 47 subjects were analyzed for the effect of LC-Plasma on salivary shedding of HHV-6 and HHV-7 DNA, respectively (Figure 2).

### 3.2. Baseline of HHV-6 and HHV-7 Shedding Patterns 

Of the 54 subjects included, 47 (87%) excreted HHV-6 DNA in saliva at least once during the first period, and 53 (98%) of the 54 subjects excreted HHV-7 DNA in saliva at least once during the first period. To elucidate the factors controlling salivary HHV-6 and HHV-7 DNA shedding, we analyzed the associations between these viral DNA loads and sex, age, and night shifts (Table 1). No significant association was observed between night shifts and shedding of HHV-6 (*P* = 0.667) or HHV-7 (*P* = 0.234) DNA. Mean HHV-6 DNA load was significantly higher in subjects under 40 years old than in those 40 or older (median/interquartile range (IQR), 19,738/5519−45,049 vs. 6638/3231−14,869 copies/mL; *P* = 0.018). HHV-7 DNA loads were significantly higher in males than in females (median/IQR, 2,676,363/1,425,588–5,941,663 vs. 552,575/50,469–3,915,838 copies/mL; *P* = 0.018) and nonsignificantly higher in subjects under 40 years old than in those 40 or older (median/IQR, 2,676,363/436,688–6,437,419 vs. 881,688/104,869–2,808,094 copies/mL; *P* = 0.059).

### 3.3. Effect of LC-Plasma Intake on Salivary Viral DNA Shedding

To elucidate the effect of LC-Plasma on salivary shedding of viral DNA, mean HHV-6 (*n* = 43) and HHV-7 (*n* = 47) DNA loads were compared over the three periods. There were no significant differences in HHV-6 DNA loads between the first and second periods (median/IQR, 15,450/5800–32,356 vs. 14,725/6400–28,788; *P* = 0.124) or between the second and third periods (median/IQR, 14,725/6400–28,788 vs. 11,975/6288–24,250 copies/mL; *P* = 0.665) (Figure 3).

Then, because the initial analyses revealed that age and sex were associated with the levels of salivary shedding of viral DNA, subjects were divided into two subgroups by age (younger age < 40 years old or older age ≥40 years old) or sex (male or female), and subgroup analyses were carried out. In the younger age subjects (*n* = 25), mean HHV-6 DNA loads were significantly lower in the second period than in the first period (median/IQR, 22,775/9300–60,463 vs. 20,325/6525–38,000 copies/mL; *P* = 0.020) (Figure 4). Meanwhile, in older subjects regardless of sex, no statistically significant difference in HHV-6 DNA load was detected.

For HHV-7 DNA, there was no significant difference between the first and second periods (median/IQR, 2,353,867/355,850–5,504,956 vs. 1,685,975/224,288–5,447,700 copies/mL; *P* = 0.715) or between the second and third periods (median/IQR, 1,685,975/224,288–5,447,700 vs. 1,899,600/393,300–5,840,888 copies/mL; *P* = 0.624) (Figure 3). Furthermore, there were no significant differences in HHV-7 DNA load between subgroups (younger vs. older or male vs. female) (Figure 4).

## 4. Discussion

To evaluate the effect of LC-Plasma on salivary shedding of HHV-6 and 7 DNA, baseline shedding of viral DNAs was measured in the first period. As previously demonstrated [8,12,14], the levels of these viral DNAs differed among individuals, and the pattern of salivary viral DNA shedding was associated with the subject’s background. HHV-6 DNA loads were significantly higher in the subjects under 40 years old than in those 40 or older (*P* = 0.018). HHV-7 DNA loads were also higher in subjects under 40 years old, but the difference was not statistically significant (*P* = 0.059). Although no sex difference was observed in salivary HHV-6 DNA shedding (*P* = 0.170), salivary HHV-7 DNA shedding was significantly higher in males than in females (*P* = 0.018) (Table 1). This is the first report of patterns of HHV-6 and HHV-7 DNA salivary shedding according to subjects’ backgrounds in healthy adults. In a previous study, we demonstrated that healthy adults with a high frequency of HHV-7 shedding in saliva (infectious virus) were more likely to have a high viral DNA load than those with a low frequency of viral shedding [12]. Recent work suggested that active viral replication of *Macaca*
*nemestrina* herpesvirus 7, a macaque homolog of HHV-7, in the salivary gland is tightly regulated at the transcription level and correlates well with salivary viral DNA shedding [27]. In addition, the salivary tissues from the nine macaques exhibited distinct viral gene expression patterns. Together with our findings in this study, these observations suggest that replication of HHV-7 and possibly HHV-6 in the salivary gland is controlled by various host factors. Previous studies carried out in recipients of hematopoietic stem cell transplantation revealed that younger age and male sex were significant risk factors of HHV-6 reactivation in peripheral blood [28,29]. This study also suggested that younger age is associated with increased replication of HHV-6 and HHV-7 in the salivary gland. Cortisol and dehydroepiandrosterone belong to glucocorticoid steroid hormones. They are released from the adrenal glands in response to stress. Cortisol play role in anti-inflammatory and immunosuppressive, and dehydroepiandrosterone is an antagonist to cortisol. The imbalance among these hormones may develop host immune suppression. In addition, the dehydroepiandrosterone level steadily decreases with age. Therefore, herpesvirus could theoretically be reactivated in older people due to stronger immune suppression in the elderly. Therefore, our findings were in conflict with our hypothesis. Recently, Kobayashi et al. also reported a negative correlation between age and salivary HHV-6 DNA levels [8]. Regardless, the hypothalamus–pituitary–adrenal axis, which mediates the production of stress hormones, may play an important role in age-related changes of salivary viral shedding, as already demonstrated by analysis of astronauts during space flight [30]. Therefore, further detailed analysis is needed to elucidate age-related HHV-6 and HHV-7 reactivation in the salivary gland and its underlying mechanisms.

We did not observe a significant inhibitory effect of LC-Plasma on salivary shedding of HHV-6 and HHV-7 DNA. However, subgroup analysis showed that HHV-6 DNA load in saliva was decreased by LC-Plasma intake in younger subjects. This finding suggests that LC-Plasma is a more effective immune activator in younger people. LC-Plasma activates pDCs in vitro and in vivo [15]. Vora et al. demonstrated that proportion and absolute numbers of pDCs in peripheral blood decrease with age [31]. Therefore, the immune activation effect of LC-Plasma via pDCs may be weakened in the elderly, who have reduced numbers of pDCs. In fact, several studies demonstrating that LC-Plasma intake decreases the risk of infection and fatigue were conducted in subjects under the age of 40 [8,22,32]. In addition to subject age, a seasonal effect may have influenced our assessment of viral reactivation in this study; the duration of LC-Plasma intake (second period) corresponded to the summer season in Japan (15 July 2019−11 August 2019), and heat stress negatively affects immune response [33,34]. Furthermore, pDC activity declines in the summer [23]. Therefore, the inhibitory effect of LC-Plasma on viral reactivation may have been underestimated. To determine the exact effect of LC-Plasma on viral reactivation, a differently designed study should be conducted to exclude seasonal effects in younger subjects. 

Although the mechanism of HHV-6 and HHV-7 reactivations in the salivary glands has not been clarified, salivary shedding of these viruses has been associated with fatigue and depression [14,35]. Meanwhile, several studies have suggested that LC-Plasma intake can decrease fatigue during consecutive high-intensity exercise sessions in young male athletes [22] and improve work performance [21]. In addition to the effect of LC-Plasma on viral reactivation, analysis of its effects on physical and mental changes is also important for future experiments.

## 5. Conclusions

This is the first study to show the characteristics of salivary shedding of HHV-6 and HHV-7 DNA in healthy adults, and that LC-Plasma decreases HHV-6 reactivation in the salivary gland in younger adults.

## Figures and Tables

**Figure 1 microorganisms-09-01683-f001:**
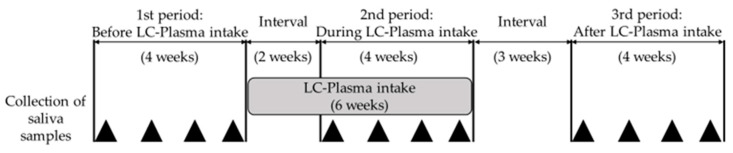
Study design: Four saliva samples were collected during each period, and HHV-6 and HHV-7 DNA loads were measured by real-time PCR. After the 4-week first period, all subjects took *Lactococcus lactis* strain Plasma (LC-Plasma) for 6 weeks. Two weeks after initiation of LC-Plasma intake (the period required for expression of effects), the subjects entered the second period. After a 3-week washout period, the subjects entered the third period. The black triangles indicate the time points when saliva samples were collected from the subjects.

**Figure 2 microorganisms-09-01683-f002:**
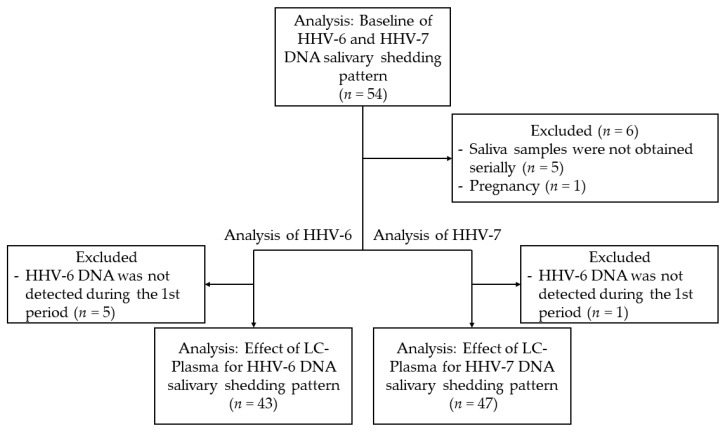
Flowchart of the selection process: Initially, 54 volunteers were enrolled in this study, and all subjects were analyzed for baseline HHV-6 and HHV-7 DNA salivary shedding patterns. Subjects who did not meet the inclusion criteria were excluded from the analysis of the effects of LC-Plasma on HHV-6 and HHV-7 DNA salivary shedding patterns.

**Figure 3 microorganisms-09-01683-f003:**
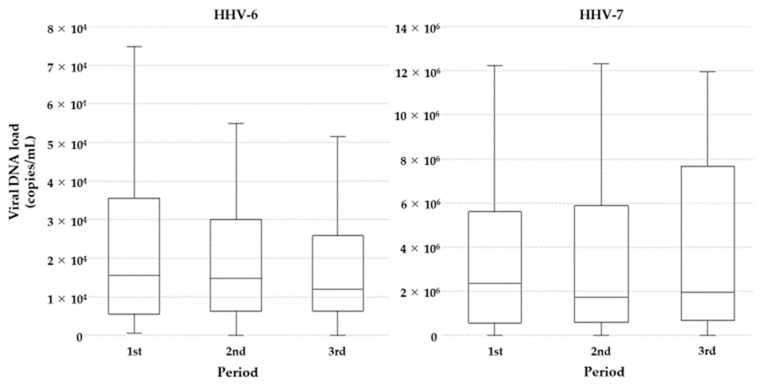
Effect of LC-Plasma on salivary viral DNA shedding: Box plots describe the comparison of HHV-6 and HHV-7 DNA loads over the three periods. Boxes and lines within the boxes represent the first through third quartiles and median, respectively. The lines outside the boxes represent the minimum and maximum values (excluding outliers).

**Figure 4 microorganisms-09-01683-f004:**
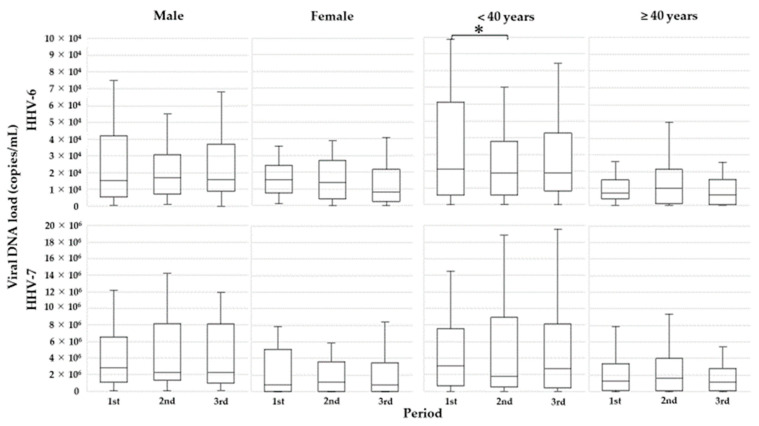
Comparison of mean HHV-6 and HHV-7 DNA loads over the three periods, by subgroup. *: Significantly different between the first and second periods.

**Table 1 microorganisms-09-01683-t001:** Baselines of HHV-6 and HHV-7 DNA salivary shedding patterns.

			HHV-6		HHV-7	
		Overall *n* (%)	DNA Load (Copies/mL) Median/SD	*p*-Value	DNA Load (Copies/mL) Median/SD	*p*-Value
**Sex**	**Male**	27 (50.0)	14,888/5494–32,356	0.170	2,676,363/1,425,588–5,941,663	0.018
	**Female**	27 (50.0)	9300/582–18,925		552,575/50,469–3,915,838	
**Age**	**<40 years**	31 (57.4)	19,738/5519–45,049	0.018	2,676,363/436,688–6,437,419	0.059
	**≥40 years**	23 (42.6)	6638/3231–14,869		881,688/104,869–2,808,094	
**Night shift**	**Yes**	20 (37.0)	14,369/3391–30,722	0.667	2,656,950/1,311,269–5,520,622	0.234
	**No**	34 (63.0)	9506/3850–22,609		1,045,719/44,197–4,445,647	

## Data Availability

Not applicable.

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
