# Peer review of "Effect of *Lactococcus lactis* Strain Plasma on HHV-6 and HHV-7 Shedding in Saliva: A Prospective Observational Study"

_microorganisms, 2021, doi:10.3390/microorganisms9081683_

Round 1
Reviewer 1 Report
Given that HHV-6 and HHV-7 are widespread and persist in the human body throughout life, and that these viruses can reactivate, for example, during prolonged stress or illness, it is important to study drugs that can reduce viral load or even avoid reactivation.
The manuscript is easy to understand and the design of the study is sufficiently understandable, but several clarifications are needed:
- Given that the taxonomy of HHV-6 has changed and that HHV-6A and HHV-6B are currently separate viruses, please explain whether you have evaluated HHV-6B and HHV-6A in your study. In the introduction, you refer to studies that show that HHV-6B is most commonly found in saliva, so it would be important to understand which of these viruses you find most often in your study. Please include this information in the manuscript.
- Were study participants instructed on the use of LC-Plasma: morning or evening, before or after meals? Are these aspects relevant to the specific study? Please describe.
- In the section of methods it is necessary to describe which positive and negative HHV-6 and HHV-7 controls were used in real-time PCR reactions.
How did you check the quality of the isolated DNA, whether in-house genes were used in parallel? - Section-Results, Line 135: For the following reasons, eleven and seven subjects were.....The sentence should be rewritten to make it clear what the figures 7 and 11 refer to.
- Line 136:....saliva samples were not serially collecting....Grammar error.
- Figure 2. An error in indicating excluded persons on HHV-7 needs to be corrected. At present, HHV-6 is written instead of HHV-7 as it should be and the number of excluded persons is incorrect.
- Line 143 ...Subjects who did not meet the inclusion criteria were excluded from the analysis of the effects of LC-Plasma on HHV-7 and HHV-6? DNA salivary shedding pattern.
Reviewer 2 Report
The authors describe in this original articles a crosstalk of HHV-6/7 viral shedding and Lactococcus lactis strain plasma (LC-plasma) in observational study. They focused on LC-Plasma intake that it decreased and increased HHV-6/7 reactivation in the salivary gland. In this study, 54 health individuals were enrolled. The authors showed the viral load by quantitative PCR (qPCR) investigating the U31 tegument gene. The authors described well the technical method used in this study.
In my opinion, the article is suitable in Microorganisms publication. However, the manuscript need minor revision.
Minor points.
Q1. The authors must describe in Material and Methods section the qPCR. They focus on the positive control used to check the DNA genome copy/ml. They also described the housekeeping probes.
Q2. In this study, it was analyzed the HHV-6/7 viral shedding. Do they know the crosstalk with Epstein-Barr (EBV) lytic infection? EBV infects the 95% of worldwide populations. Several stimulus induces in vivo a productive lytic phase related to viral release. Did you focus on it? They should perform a qPCR on EBV gene such as BALF5 gene. They could use as positive control Namalwa cell line.
Q3. Add several references about query 2.
Q4. The authors must underline in the study the viral shedding relating to job types. It is known that several stress can increase or decrease the herpesviruses release. It could be a good point of view.
Described it.

Author Response
Response to Reviewer 2 Comments
The authors describe in this original articles a crosstalk of HHV-6/7 viral shedding and Lactococcus lactis strain plasma (LC-plasma) in observational study. They focused on LC-Plasma intake that it decreased and increased HHV-6/7 reactivation in the salivary gland. In this study, 54 health individuals were enrolled. The authors showed the viral load by quantitative PCR (qPCR) investigating the U31 tegument gene. The authors described well the technical method used in this study.
In my opinion, the article is suitable in Microorganisms publication. However, the manuscript need minor revision.
Comments:
Q1. The authors must describe in Material and Methods section the qPCR. They focus on the positive control used to check the DNA genome copy/ml. They also described the housekeeping probes.
Response 1: As suggested by the reviewer, information of positive controls for making standard curves was added in the revised manuscript. (Line116). Housekeeping gene was not measured in this study.
Q2. In this study, it was analyzed the HHV-6/7 viral shedding. Do they know the crosstalk with Epstein-Barr (EBV) lytic infection? EBV infects the 95% of worldwide populations. Several stimulus induces in vivo a productive lytic phase related to viral release. Did you focus on it? They should perform a qPCR on EBV gene such as BALF5 gene. They could use as positive control Namalwa cell line.
Q3. Add several references about query 2.
Response 2, 3: Thank you for your fruitful comment. Since we focused on HHV-6 and HV-7 infection, we did not carry out qPCR on EBV gene in this study. I am going to analyze an association between LC-plasma administration and EBV shedding.
Q4. The authors must underline in the study the viral shedding relating to job types. It is known that several stress can increase or decrease the herpesviruses release. It could be a good point of view.
Response 4: Thank you for your suggestion. I agree with your comment. However, since number of study subject was insufficient for evaluation of the impact of job types on HHV-6 and -7 DNA shedding, we did not perform such analysis. We would like to examine the impact of job types on each viral DNA shedding in future